# Differential scanning fluorimetric analysis of the amino-acid binding to taste receptor using a model receptor protein, the ligand-binding domain of fish T1r2a/T1r3

**Takashi Yoshida[1], Norihisa Yasui[1], Yuko Kusakabe[2], Chiaki Ito[1], Miki Akamatsu[3], Atsuko Yamashita[1]** *

**1** Graduate School of Medicine, Dentistry and Pharmaceutical Sciences, Okayama University, Okayama, Okayama, Japan, **2** Food Research Institute, National Agriculture and Food Research Organization, Tsukuba, Ibaraki, Japan, **3** Graduate School of Agriculture, Kyoto University, Kyoto, Kyoto, Japan

* a_yama@okayama-u.ac.jp

**Data Availability Statement:** All relevant data are within the manuscript and its Supporting Information files.

## Abstract

Taste receptor type 1 (T1r) is responsible for the perception of essential nutrients, such as sugars and amino acids, and evoking sweet and umami (savory) taste sensations. T1r receptors recognize many of the taste substances at their extracellular ligand-binding domains (LBDs). In order to detect a wide array of taste substances in the environment, T1r receptors often possess broad ligand specificities. However, the entire ranges of chemical spaces and their binding characteristics to any T1rLBDs have not been extensively analyzed. In this study, we exploited the differential scanning fluorimetry (DSF) to medaka T1r2a/T1r3LBD, a current sole T1rLBD heterodimer amenable for recombinant preparation, and analyzed their thermal stabilization by adding various amino acids. The assay showed that the agonist amino acids induced thermal stabilization and shifted the melting temperatures ($T_m$) of the protein. An agreement between the DSF results and the previous biophysical assay was observed, suggesting that DSF can detect ligand binding at the orthosteric-binding site in T1r2a/T1r3LBD. The assay further demonstrated that most of the tested L-amino acids, but no D-amino acid, induced $T_m$ shifts of T1r2a/T1r3LBD, indicating the broad L-amino acid specificities of the proteins probably with several different manners of recognition. The $T_m$ shifts by each amino acid also showed a fair correlation with the responses exhibited by the full-length receptor, verifying the broad amino-acid binding profiles at the orthosteric site in LBD observed by DSF.

## Introduction

Taste perception starts with specific molecular interactions between taste substances and taste receptors in the oral cavity. Various chemicals evoking taste sensation are categorized into five basic taste modalities and perceived by distinct receptors specialized to each modality [1, 2]. Among the five modalities, sweet, umami, and salty tastes are generally recognized as

**Funding:** This work is supported by Grant-in-Aid from the Ministry of Education, Culture, Sports, Science and Technology (MEXT), Japan (Grant Number 18H04621 and 17H03644 to AY), and Japan Society for the Promotion of Science (JSPS), Japan (Grant Number 15H05370 to NY). The funders had no role in study design, data collection and analysis, decision to publish, or preparation of the manuscript.

**Competing interests:** The authors have declared that no competing interests exist.

preferable tastes and induce positive hedonic responses, while bitter and sour tastes primitively induce negative hedonic responses to animals, including humans [3].

Among the preferable taste modalities, sweetness and umami are perceived by taste receptor type 1 (T1r) proteins conserved among vertebrates [4]. T1rs are class C G protein-coupled receptors (GPCRs) [5], which commonly function as homo- or heterodimeric receptors [6]. Specifically, in mammals, T1r2/T1r3 heterodimer serves as a sweet taste receptor, while T1r1/T1r3 heterodimer serves as an umami taste receptor [7–9]. These receptors recognize major taste substances by the ligand binding domains (LBDs) located at the extracellular region [10]. T1r LBDs share an architecture known as the Venus flytrap module (VFTM) characteristic to the extracellular domains of class C GPCRs, and taste substances bind to the cleft between the bilobal subdomains composing the VFTM (Fig 1A) [11].

Notably, in T1rLBDs, the ligand-binding sites, referred to as the orthosteric binding sites, need to accommodate taste substances covering the most part of the chemical spaces presenting the taste modality, because a single kind of receptor is responsible for the perception of a single modality. Indeed, the orthosteric binding sites in many T1rLBDs bind a wide array of chemicals; the site in human T1r2/T1r3 sweet receptor binds various mono- to oligosaccharides as glucose, fructose, and sucrose, and artificial sweeteners as dipeptide derivatives (aspartame, neotame) or sultames (Acesulfame-K, saccharin) [10], while those in mouse T1r1/T1r3 and some of the fish T1rs bind a wide array of amino acids [9, 12, 13]. The broad ligand-binding capabilities of the orthosteric sites in T1rs contrast with those in other class C GPCRs, such as metabotropic glutamate receptors or γ-aminobutyric acid (GABA) receptor B, which are more or less specific to their intrinsic agonist molecules, glutamate or GABA, respectively [14, 15].

The details of molecular interactions between T1rs and taste substances have long been unknown, due to the lack of structural information of T1rLBDs. The T1rLBD heterodimers, including human proteins, are difficult for recombinant expression and large-scale preparation [16], hampering the structural analyses. Recently, by extensive expression screening

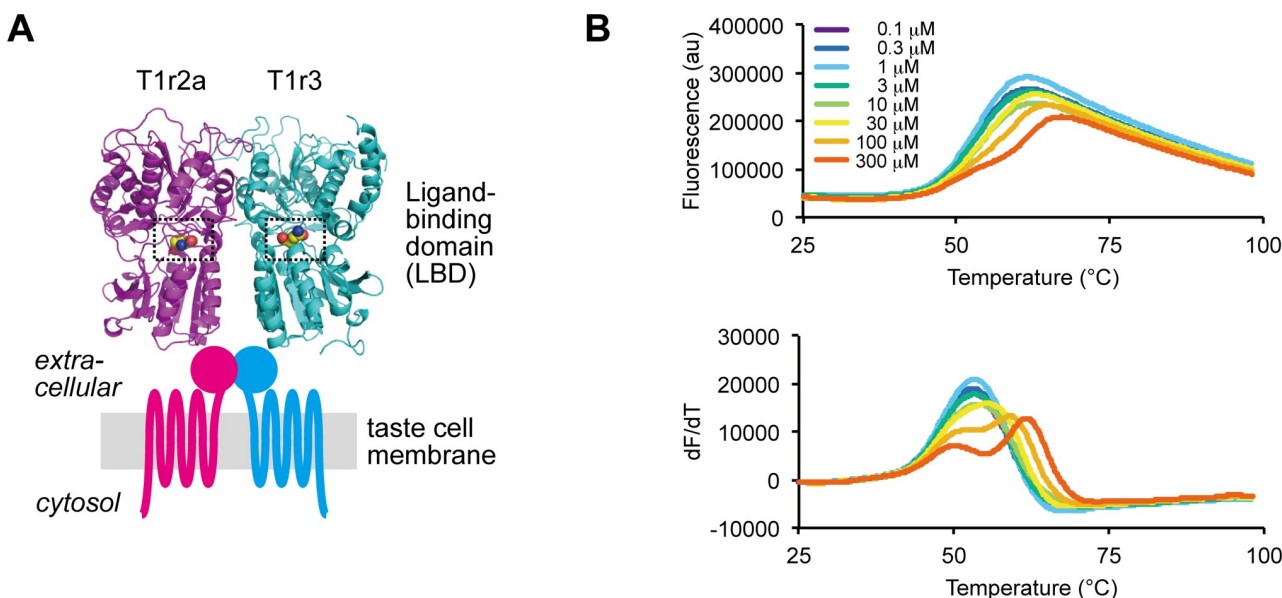

**Fig 1. Amino acid binding to medaka T1r2a/T1r3LBD.** (A) Crystallographic structure of medaka T1r2a/T1r3LBD in complex with L-glutamine (PDB ID: 5X2M) and a schematic drawing of the entire T1r receptor. The orthosteric binding sites in T1r2a and T1r3 are highlighted with dashed boxes. (B) Thermal melt curves of T1r2a/T1r3LBD (top) and their derivatives (bottom) in the presence of 0.1 ~ 300 μM L-glutamine, measured by DSF.

among vertebrate T1rLBDs, we solved the first crystallographic structures of the heterodimeric LBDs of T1r2-subtype a (T1r2a)/T1r3 LBD from medaka fish, *O. latipes*, an amino acid-taste receptor (Fig 1A) [11]. In the crystallographic structures, binding of taste-substance amino acids was observed at the orthosteric binding sites in T1r2a/T1r3LBD. The binding sites indeed possess favorable structural characteristics to accommodate various amino acids, such as a large space covered with a surface mosaically presenting negatively, positively and uncharged regions. Nevertheless, the entire ranges of chemical spaces and their binding characteristics to the orthosteric sites in T1r2a/T1r3, as well as any other T1rs, have not been extensively analyzed. So far, we have employed two kinds of methodologies: isothermal titration calorimetry for direct measurement of the binding heat generated by interactions between the T1rLBD protein and a taste substance; and a Förster resonance energy transfer (FRET) analysis using the T1rLBD-fluorescent protein fusions for indirect measurement of the conformational change of the protein accompanied by ligand binding [17]. However, the two methods are sample and time consuming, and only five amino acids were so far subjected to structural and biophysical analyses to examine interactions with T1r2a/T1r3LBD. In order for the extensive ligand binding analyses of the protein, an assay method with higher throughput is required.

In this study, we employed a thermal shift assay analyzed by differential scanning fluorimetry (DSF) for ligand binding analysis of T1r2a/T1r3LBD [18]. The DSF measures a thermal unfolding of a protein by detecting the change of fluorescence intensity of an environmentally-sensitive fluorescence dye binding to hydrophobic regions of the protein exposed to the solvent during its denaturation [19, 20]. Because a ligand binding to the protein generally changes its thermal stability, DSF is applicable to a ligand-binding assay. Among various assay methodologies, DSF can serve as a high-throughput method since it requires a small amount of protein for a measurement ($\sim 1$ μg), and multiple parallel measurements are feasible by the use of conventional real-time PCR equipment. The results in this study showed that the binding of the agonist amino acids induced thermal stabilization of T1r2a/T1r3LBD, which can be detected by DSF, indicating that the method can serve as a high-throughput ligand binding assay for T1rLBDs. The DSF displayed that a wide array of L-amino acids bind to the orthosteric site in T1r2a/T1r3LBD, regardless of their physicochemical properties.

## Materials and methods

### Sample preparation

The protein sample was prepared as described previously [11, 17]. Briefly, *Drosophila* S2 cells (Invitrogen) stably expressing C-terminal FLAG-tagged T1r2aLBD and T1r3LBD [11, 21] were cultured in ExpressFiveSFM (LifeTechnologies) for five days at 27 ˚C. The T1r2a/T1r3-LBD protein was purified from the culture medium by the use of ANTI-FLAG M2 Affinity Gel (SIGMA). The purified protein was dialyzed against the assay buffer (20 mM Tris-HCl, 300 mM NaCl, 2 mM CaCl$_2$, pH 8.0).

### Differential scanning fluorimetry

The protein sample ($\sim$1 μg) was mixed with Protein Thermal Shift Dye (Applied Biosystems) and 10~10,000 μM concentration of each amino acid in 20 μL of assay buffer and loaded to a MicroAmpR Fast Optical 48-Well Reaction Plate (Applied Biosystems). Specifically, to minimize the changes of the chemical and optical condition, the amino acid solutions were prepared by the exact same buffer used for the final dialysis of the protein, followed by the re-adjustment of the pH to $\sim$ 8.0 by addition of either NaOH or HCl, and mixed with the protein diluted with the same buffer.

Fluorescent intensity was measured by the StepOne Real-Time PCR System (Applied Biosystems). The temperature was raised from 25 °C to 99 °C with a velocity of 0.022 °C/sec. The reporter and quencher for detection were set as "ROX" and "none", respectively. Apparent melting transition temperature ($T_m$) was determined by the use of the maximum of the derivatives of the melt curve (dFluorescence/dT) by Protein Thermal Shift Software version 1.3 (Applied Biosystems).

We confirmed that the sample in a different buffer system (e.g. HEPES) gave consistent results with those under the condition in this study for several representative ligands (S1 Fig).

## Data analysis

The apparent dissociation constant ($K_{d\text{-app}}$) derived from the DSF results was estimated based on Eq 1 proposed by Schellman [22], assuming that the unfolding of the protein is reversible:

$$\Delta T_m = T_m - T_0 = \frac{T_m T_0 R}{\Delta H^0} ln(1 + \frac{[L]}{K_{d-app}}) \tag{1}$$

where $[L]$ is the ligand concentration; $T_m$ and $T_0$ are the apparent melting transition temperatures in the presence and absence of the ligand; $R$ is the gas constant; $\Delta H^0$ is the enthalpy of unfolding at $T_0$, assuming that there are no significant variations under the tested conditions. If the melt curves show biphasic profiles, the second (or the right side) $T_m$ values were adopted for calculation, as described in the Results section.

For multiple regression analyses shown in Fig 2B, the apparent $T_m$ values determined at different ligand concentrations were fitted to Eq 1 by using KaleidaGraph (Synergy Software), assuming that the change of the dissociation constant accompanied by the $T_m$ shift is negligible. For fitting, $T_0$ was fixed at 326.2 K, the experimentally determined value by DSF in the same experimental set (s.e.m. 0.2 K, $n = 7$), and $K_{d\text{-app}}$ and $\Delta H^0$ values were set as variables.

For S1 Table, $K_{d\text{-app}}$ was estimated using the apparent $T_m$ values determined at a single ligand concentration by substituting $T_0$ and $\Delta H^0$ in Eq 1 with 326.1 K, determined at the same experimental set (s.e.m. 0.1 K, $n = 20$), and 72.1 kcal mol$^{-1}$, the average of the fitted values of the multiple regression analyses described above (s.e.m. 7.2 kcal mol$^{-1}$, $n = 5$), respectively. The derived $K_{d\text{-app}}$ values for L-glutamine, alanine, arginine, glutamate, and glycine were found to show good agreement with those determined by FRET, within 0.55 ~ 1.55 fold of the FRET EC$_{50}$ values, if they were determined using the $\Delta T_m$ values in the range of 6 ~ 11 K (S1 Table). On the other hand, $\Delta T_m$ below 1 K or above 11 K resulted in larger deviations, such as below 0.5 fold or above 3 fold of the EC$_{50}$ values. Because $\Delta T_m$ values for most amino acids at 10 mM concentration were observed in the range of 2 ~ 11 K, $K_{d\text{-app}}$ values derived from the results at 10 mM were used for the further analysis, with the following exceptions. For L-alanine and L-glutamine, the results at 1 mM and 0.1 mM were adopted, because $\Delta T_m$ values were observed in the range of 6 ~ 11 K and the resulted $K_{d\text{-app}}$ values showed the closer agreement with the FRET EC$_{50}$ values compared to the results at 10 mM. The amino acids indicating the thermal destabilization, L-lysine and D-alanine, were not included in the further analyses.

The relationship between the side chain structures and p$K_{d\text{-app}}$ (= log $1/K_{d\text{-app}}$) values for 15 L-amino acids, excluding L-proline, was quantitatively analyzed using the classical quantitative structure-affinity relationships (QSAR) technique [23]. Classical QSAR analyses were performed using QREG ver. 2.05 [24]. The physicochemical parameters of amino acid $\alpha$-substituent groups used for the analysis were listed in S3 Table.

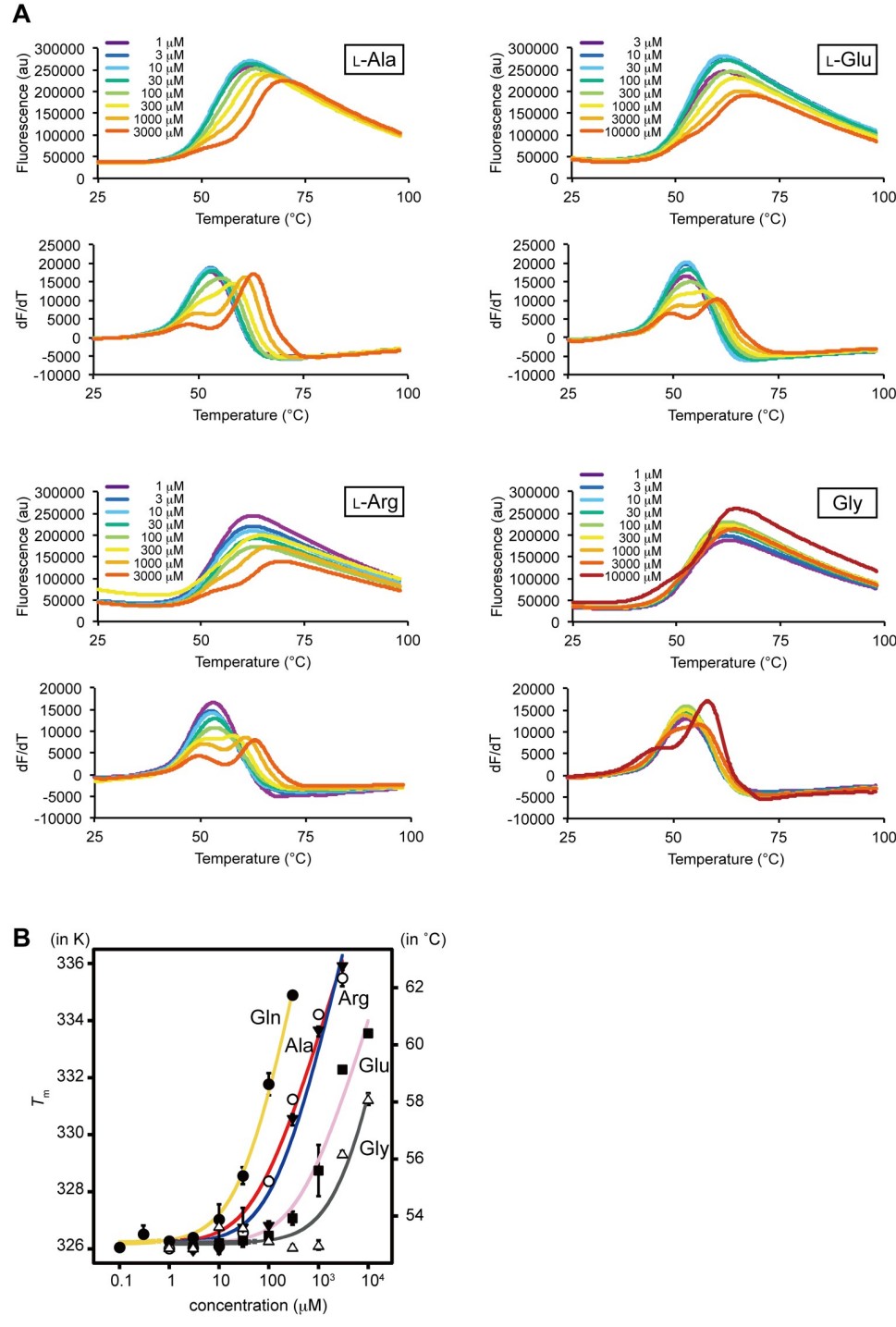

**Fig 2. Dose-dependent $T_m$ changes of T1r2a/T1r3LBD by the addition of amino acids.** (A) Thermal melt curves of T1r2a/T1r3LBD and their derivatives in the presence of 1 ~ 10,000 μM concentrations of L-alanine, arginine, glutamate, and glycine, measured by DSF. (B) Dose-dependent $T_m$ changes of T1r2a/T1r3LBD by addition of L-glutamine, alanine, arginine, glutamate, and glycine. Six technical replicates for L-glutamine and 4 technical replicates for the others were averaged and fitted to Eq 1 in Materials and Methods. Error bars, s.e.m.

## Receptor response assay

The $Ca^{2+}$-flux assay was performed using Flip-In 293 cell line (Life Technologies) stably expressing full-length T1r2a, T1r3, and $G\alpha16$-gust44 as described previously [11, 17]. The response stimulated by either 5 or 10 mM amino acid was represented as $\Delta RFU$ (delta relative fluorescence unit) defined as the maximum fluorescence intensity induced by the addition of the amino acid, subtracted with that of an assay buffer in the absence of amino acid. The estimated $EC_{50}$ values, $EC_{50\text{-est}}$, were calculated using the Hill equation as follows:

$$\Delta RFU = \frac{\Delta RFU_{max} \times [L]}{EC_{50-est} + [L]}$$

where $[L]$, $\Delta RFU$, $\Delta RFU_{max}$ were substituted by either 5 or 10 mM, $\Delta RFU$ values at 5 or 10 mM, and 104.3, the maximum $\Delta RFU$ value observed in the same set of experiments (by addition of 5 mM L-glutamine, a saturated concentration observed in previous studies [11, 17]; s.e. m. 4.24, $n = 12$). $pEC_{50\text{-app}}$ (= log $1/EC_{50\text{-est}}$) values for 13 or 15 amino acids, estimated from the results at 5 mM or 10 mM results, respectively, by excluding those giving negative $\Delta RFU$ values, were compared with $pK_{d\text{-app}}$ values.

## Results

### T1r2a/T1r3LBD exhibited thermal stabilization by binding a taste substance amino acid

An essential prerequisite for DSF application to a ligand binding assay is that the protein should show a shift of thermal melt curves accompanied by the ligand addition, *i.e.*, the protein should be either thermal stabilized or destabilized by ligand binding. In order to examine whether DSF is applicable to ligand binding analysis of T1r2a/T1r3LBD, we analyzed its thermal melt curves with various concentrations of L-glutamine, the amino acid taste substance to medaka T1r2a/T1r3LBD with the highest affinity to the protein so far analyzed [11].

T1r2a/T1r3LBD showed a thermal melt curve with a monophasic transition, with a single maximum in its derivatives, in the absence of amino acids (Fig 1B). The transition temperature of melting ($T_m$) was determined by the derivative of the melt curve and estimated as 53.0 ± 0.07 ˚C. The addition of L-glutamine shifted the melt curves toward the higher temperature side and changed the curve profiles with apparently biphasic transitions, with two maxima in their derivatives. In the biphasic melt curves in the presence of L-glutamine, the higher concentration of the ligand added, the higher temperature shifts were observed at the second (or the right side) $T_m$, as the increase of $T_m$ ($\Delta T_m$) of 8.7 ± 0.1 ˚C in the presence of 300 μM L-glutamine, while the first (or the left side) $T_m$ was observed as about 50 ˚C and did not exhibit clear thermal shifts. The results indicated that a taste-substance amino acid binding to T1r2a/T1r3LBD induces the thermal stabilization of the protein, at least at the structural portion showing the melting transition at a higher temperature side observed at the second $T_m$.

### DSF results displayed the binding of taste substance amino acid at the orthosteric sites in T1r2a/T1r3LBD

Agonist-binding to the orthosteric sites in class C GPCRs is known to induce the conformational change of LBDs, either or both of the cleft closure of the VFTM architecture within a subunit or the dimer rearrangement [14]. These conformational changes are considered to induce receptor activation [25]. The crystallographic analyses of medaka T1r2a/T1r3LBD displayed that L-glutamine, alanine, arginine, glutamate, and glycine bind to the orthosteric sites [11], and the binding actually induced the conformational change of the protein as judged by

Table 1. $K_{d\text{-}app}$ and EC$_{50}$ values for the amino-acid binding to T1r2a/T1r3LBD estimated by different biophysical methods.

| Amino acid | DSF $K_{d\text{-}app}$ (μM)[†] | FRET EC$_{50}$ (μM)[‡] |
|---|---|---|
| L-Gln | 30.9 ± 5.8 | 11.5 ± 3.4 |
| L-Ala | 54.1 ± 24.5 | 141 ± 37 |
| L-Arg | 131 ± 66 | 190 ± 35 |
| L-Glu | 422 ± 211 | 1070 ± 382 |
| Gly | 3570 ± 4090 | 6180 ± 3320 |

[†]The values are fitted parameters ± s.e. to the equation curves reported in Schellman [22]. Six technical replicates for L-glutamine and 4 technical replicates for the others were averaged and used for fitting.

[‡]The values are reported in Nuemket, Yasui, *et al.* [11].

FRET changes in accordance with the addition of the ligands [17]. In order to verify whether the $T_m$ shift observed by DSF monitors the ligand binding at the orthosteric sites, we compared the DSF results in the presence of the above five amino acids with the reported results analyzed by the FRET measurement.

All five amino acids previously confirmed the binding to T1r2a/T1r3LBD induced the thermal stabilization of the protein, with changing the melt curve profiles as biphasic transitions (Fig 2A). We plotted the $T_m$ values (if the melt curves are biphasic, the second $T_m$ values as described above) in the presence of 8 or 9 different concentrations of amino acid in Fig 2B. For comparison with the previous FRET results, the apparent dissociation constant ($K_{d\text{-}app}$) for each amino acid was estimated using a simple thermodynamic model [22] (Table 1). The $K_{d\text{-}app}$ values determined by DSF showed fair agreement with EC$_{50}$ values for the FRET changes with the addition of the amino acids. The results suggest that the thermal stabilization of T1r2a/T1r3LBD by the addition of amino acids detected by DSF is attributed to the ligand bindings at the orthosteric sites.

## T1r2a/T1r3LBD has a broad L-amino acid binding profile irrespective of the physicochemical properties of their α-substituent groups

We extended the DSF analysis to the other amino acids to explore the ligand specificity of T1r2a/T1r3LBD. Most of the L-amino acids tested induced the shifts of $T_m$ toward the higher temperatures (Fig 3 and S2 Fig). A wide array of L-amino acids, with various physicochemical properties in terms of size, hydrophobicity/hydrophilicity, and charge, induced thermal stabilization of the protein. The results clearly indicate the broad specificity of T1r2a/T1r3LBD to L-amino acids. There are only two exceptions among those tested, L-aspartate and lysine, which shifted the melt curves toward the lower temperature side (S2 Fig), thereby suggesting the thermal destabilization of the protein.

In contrast to binding abilities of L-amino acids to T1r2a/T1r3, a representative D-amino acid, D-alanine, did not induce a significant $T_m$ shift by adding up to 10 mM, despite the fact that its enantiomer L-alanine exhibited large $T_m$ shifts (Figs 2 and 3). These results indicate that the protein has specificity to L-amino acids, as observed on the conformation changes of LBD indicated by FRET changes [11].

In order to verify the amino acid binding profiles of T1r2a/T1r3LBD observed by DSF described above, the results were compared with the response assay using the full-length receptor. The T1r2a/T1r3 receptor from *O. latipes* reportedly responds to a wide array of L-amino acids [12]. We confirmed the broad specificity on L-amino acid responses of this receptor by

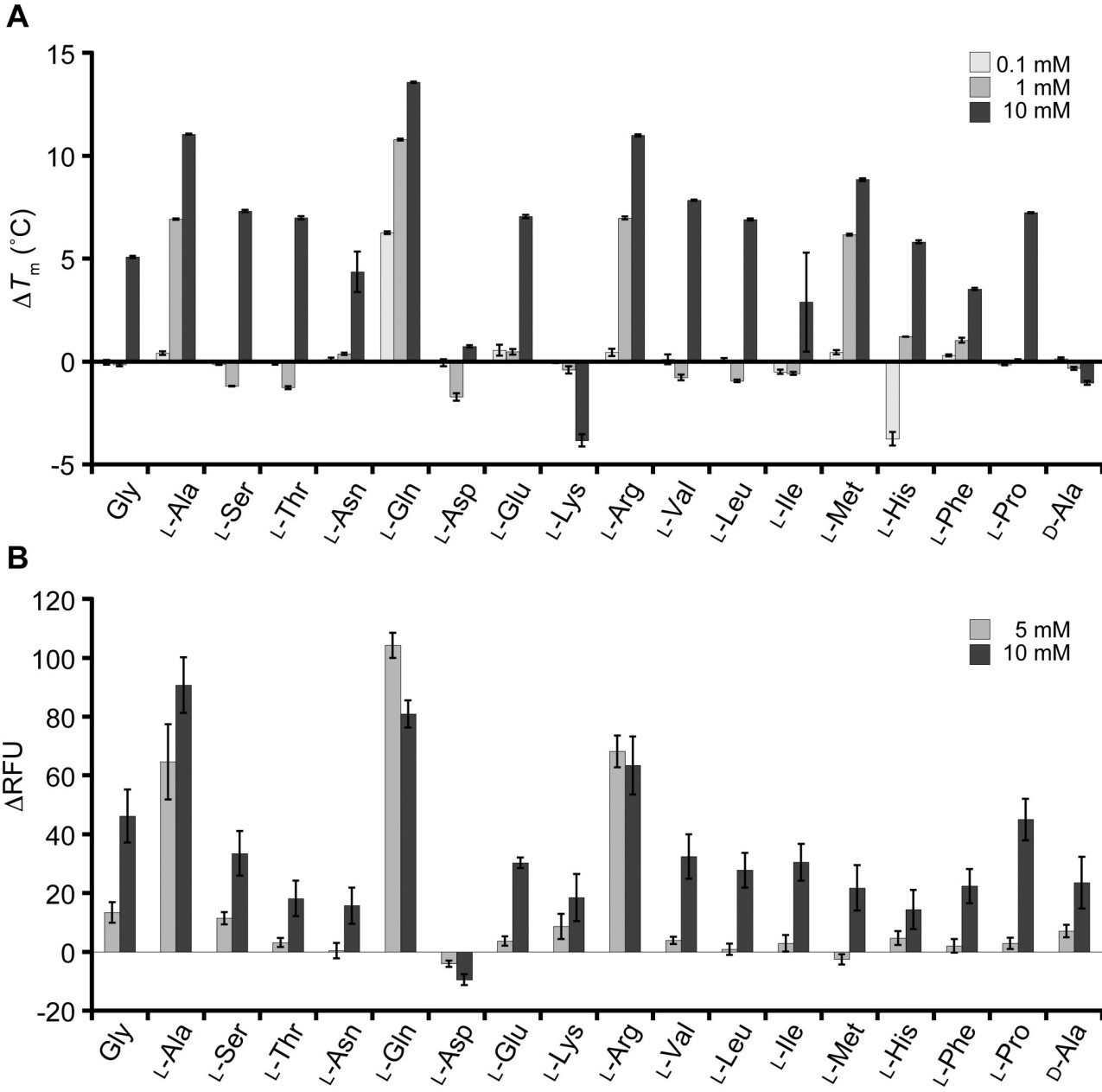

**Fig 3. Amino-acid binding profiles of T1r2a/T1r3LBD, analyzed by DSF.** (A) Thermal stabilization of T1r2a/T1r3LBD by the addition of various amino acids. Average $\Delta T_m$ in the presence of 0.1, 1, and 10 mM of each amino acid are shown. Error bars, s.e.m. ($n = 4$). (B) Responses of the T1r2a/T1r3 full-length receptor to various amino acids in 5 or 10 mM concentration monitored as an elevation of intracellular $Ca^{2+}$ elevation. The average $\Delta$RFU (difference in fluorescence intensity of the calcium indicator) and s.e.m. of 6 technical replicates for each amino acid are shown.

use of the same gene clones used for the DSF analyses (AB925918 and AB925919; Fig 3B and S2 Table). In contrast, D-alanine induced significantly weak responses compared to its enantiomer L-alanine (Fig 3B), as D-glutamine reported previously [11].

Because of the limitation of the experimental system, which does not allow full exploration to high amino-acid concentrations to determine the $EC_{50}$ values of low-affinity ligands [11], the relationships between the DSF results and the response assay results were assessed by use of a couple of alternative parameters. If we compared the observed $T_m$ shifts ($\Delta T_m$) of the LBD

at 10 mM amino acid analyzed by DSF with the observed responses (ΔRFU) by addition of the same ligand concentration, both values showed a moderate positive correlation ($n$ = 18, $r$ = 0.700; S3 Fig). In addition, we estimated the binding affinities and the potencies of the receptor responses from the DSF and the response assay results at a single concentration, respectively (S1 and S2 Tables), and confirmed that the p-scaled values of both also showed a moderate positive correlation ($n$ = 15, $r$ = 0.769 or $n$ = 13, $r$ = 0.748; S3 Fig). These results indicated the correlation between the amino-acid binding profiles of T1r2a/T1r3LBD observed by DSF and the receptor response profiles of the full-length T1r2a/T1r3 and confirmed the broad amino-acid specificity of this protein.

In the DSF analyses, while most of the L-amino-acids induced thermal stabilization of T1r2a/T2r3LBD, the extent of $T_m$ shifts of each amino acid was varied, suggesting their different affinities to the protein. In order to assess whether there are any determinant chemical properties for the affinity to the protein, classical QSAR of amino acids were performed. The relationship between the $K_{d\text{-app}}$ values, determined above, with various parameters used in classical QSAR, such as hydrophobicity, hydration, polarity, hydropathy, charge, and volume of the substituent groups, was inspected (S1 and S3 Tables). However, as far as analyzed, no equation showing a significant correlation with the affinities to T1r2a/T1r3LBD was obtained. The result suggests that the amino acid specificity of T1r2a/T1r3LBD is unlikely governed by a single or a combination of some physicochemical properties of a ligand but could be affected by multiple structural and physicochemical factors of both the protein and the ligand.

## Discussion

Chemosensory receptors, including taste receptors, are required to recognize a wide array of chemicals in the environment. The crystal structure of T1r2a/T1r3LBD from *O. latipes* showed that the orthosteric ligand-binding pockets shared favorable structural characteristics to accommodate various amino acids [11]. In this study, we first verified a correlation between the ligand-induced thermal stabilization of T1r2a/T1r3LBD analyzed by DSF and the ligand binding to the orthosteric site at the LBD. Furthermore, we showed a broad amino acid spectrum of the binding capability by T1r2a/T1r3LBD. Consistent with the previous knowledge about class C GPCR that the ligand binding at the orthosteric site induces receptor responses [6], DSF results exhibited a correlation with amino acid responses analyzed by the calcium influx assay using the full-length receptor.

### Amino acid specificity of T1r2a/T1r3LBD

The DSF results showed the differences in the extent of $T_m$ shifts induced by each amino acid, indicating their different affinities. The results suggest that the manner of recognition of the $\alpha$-substituent groups of ligand amino acids by T1r2a/T1r3LBD is not identical but varied. Indeed, it is intriguing that two pairs of basic or acidic amino acids, arginine and lysine or glutamate and aspartate, gave opposite effects to the protein; the former thermally stabilized the protein while the latter destabilized the protein (Fig 3A).

In this study, we could not find any significant quantitative relationships between the physicochemical properties of the amino acids and their affinities with the protein. This is consistent with the structural observation of the ligand binding-pocket in T1r2a/T1r3LBD: there are no apparent structural characteristics or functional groups to determine specificity to the $\alpha$-substituent groups of the bound amino acid in the protein, and the substituent groups of the different amino acids take different conformations [11]. Therefore it is likely that T1r2a/T1r3LBD has multiple different manners of recognition of the $\alpha$-substituent groups, and this property is also favorable for achieving the broad amino-acid perceptibility.

Another important structural characteristics of the ligand binding-pocket in T1r2a/T1r3LBD is that the $\alpha$-substituent groups of the bound amino acid are recognized in hydrated states, and almost all interactions between the groups and the protein are made through water molecules [11]. Similar interactions were observed on the bacterial periplasmic oligopeptide-binding protein OppA, also able to bind peptides with widely varying amino acid sequences [26]. An extensive thermodynamic analysis of OppA revealed that the peptide-protein interactions clearly showed the enthalpy-entropy compensation phenomenon [26], where the enthalpy and entropy changes by the interactions are correlated and give opposite effects on the free energy [27]. A similar phenomenon might occur on T1r-amino acid binding and could make the contributions of each physicochemical property of the ligand to the free energy obscure.

However, it should be noted that the estimations of binding affinities in this study are indirect and approximate. In addition, the reason why lysine or aspartate induced thermal destabilization is unclear. Further structural and precise interaction analyses are required to elucidate the determinant of the ligand specificity of the receptor.

## Thermodynamic properties of T1r2a/T1r3LBD

The DSF results not only provide information about the ligand binding to T1r2a/T1r3LBD, but also the thermodynamic properties of the protein itself. It is noteworthy that the protein shows biphasic melt curves, highlighted by the presence of two maxima in their derivatives, in the presence of a high concentration of amino acids (Figs 1 and 2, and S2 Fig). The profiles contrast with a previous report that human and mouse T1r2LBD, prepared as a single subunit by *E. coli* expression, showed two-state transitions between apo and ligand-bound forms by differential scanning calorimetry (DSC), indicating monophasic melting of the protein [28].

Several cases showing biphasic unfolding characteristics were reported, such as high-affinity ligand binding [29], an increase of the free ligand during the unfolding of the protein caused by the release of the ligand from the denatured protein [30], and the presence of multiple structural regions with lower and higher stabilities [31]. While the former two cases unlikely occurred on T1r2a/T1r3LBD, because the biphasic features in those cases were observed at low concentrations of the ligands, the last case might conform with this protein.

T1r2a/T1r3LBD is composed of multiple structural elements, potentially showing different thermal stabilities: individual subunits, T1r2a and T1r3, which further consist of two subdomains LB1 and LB2, with the orthosteric amino-acid binding sites in between the subdomains, and the dimerization of the two subunits through intermolecular interaction between LB1 of each subunit, further connected by an intermolecular disulfide bond at a loop region atop the dimer [11]. The transition at the higher temperature side observed in this study, indicated as the second $T_\mathrm{m}$, likely reflected the unfolding accompanied with the destruction of the amino-acid binding site determining the receptor specificity, because the extent of $T_\mathrm{m}$ shifts correlated with the extent of the conformational change of the LBD and the receptor responses (Table 1 and Fig 3). The site is most probably the orthosteric site in T1r2aLBD because the orthosteric amino-acid binding site in T1r2a shows discriminative ligand recognition manners compared to that in T1r3, although the latter site also shares amino-acid binding capability as observed in the crystallographic analysis [11]. However, since this transition was observed as a single phase, it is difficult to distinguish the effects of the ligand binding at the sites in T1r2aLBD and T1r3LBD separately. Therefore, it is also possible that the destructions of the binding sites in the two subunits occur independently but overlapped in the analyses, or occur cooperatively.

On the other hand, because the transition at the lower temperature side did not show the thermal stabilization associated with the addition of amino acid, it is unlikely associated with

the destruction of the known amino-acid binding sites in T1r2a/T1r3LBD. We speculate that one of the candidate events related to this transition might be dimer decomposition. It has been reported that the extracellular domain of another class C GPCR, metabotropic glutamate receptor 2 dimer, is in a fast dynamic exchange between different conformational states regardless of the presence of agonist or antagonist, although the ligands change the conformational equilibriums [32], as is also observed in other GPCRs [33]. If the decomposition of the dimerization of T1r2aLBD and T1r3LBD is triggered not by a certain conformational state but by conformational exchange, then the speculation is in accord with the DSF results. The speculation is also in accord with the previous observation that a single subunit of T1r2LBD showed monophasic melting profiles [28].

### Future applicability to taste assays

From a practical point of view, this study indicates the future applicability of DSF to a quantitative assay method for taste substances that induce gustation by T1r receptors, *i.e.*, sweet and umami. Effective assay methods to evaluate taste qualities and intensities are required for basic taste research in academia as well as for new taste-substance development in food industries. Currently, taste evaluation in these industries is mainly dependent on rating by human participants. Such sensory evaluations are scientifically verified by *in vivo* animal behavior tests or *in vitro* analyses as calcium influx assays using receptor-expressing cells with cytosolic calcium indicators or biomimetic sensors specialized to the detection of taste substances in research institutes, which are equipped with special devices or facilities that are required for the analyses. Compared to these methods, protein-based binding assays are advantageous to feasibility, reproducibility, and scalability. So far, protein-based assays of T1rs were attempted by the use of single subunit T1rLBDs obtained by refolding inclusion bodies expressed in *E. coli*, and they were applied to intrinsic tryptophan fluorescence measurement, circular dichroism measurement, isothermal titration calorimetry (ITC), NMR, and DSC [28, 34, 35]. We applied T1r2a/T1r3LBD from *O. latipes*, a sole T1rLBD heterodimer protein amenable for recombinant protein preparation at present, to ITC and a FRET analysis previously [11, 17]. However, all of these methods are either sample or time consuming, and not trivial. In contrast, DSF can serve as a high-throughput binding assay by comparing the relative extent of the thermal stabilization of the protein.

However, a couple of points should be kept in mind for applying the method for an actual taste assay. The target site for some taste substances or inhibitors for T1rs, such as a sweet protein brazzein, cyclamate, and lactisole, are known to bind to the sites other than LBD of T1rs, such as transmembrane domain or the cysteine-rich domain, the downstream region of LBD at the extracellular side [10, 36, 37]. In such cases, the ligand binding is unable to be detected by DSF using LBD. In addition, since there are no known antagonists for T1r2a/T1r3 from *O. latipes*, we could not test whether agonists and antagonists can be distinguished by the use of DSF results. Various types of actions of amino acids, such as allosteric or inhibitory actions, might underlie a non-strict correlation between the ligand binding and receptor responses observed in this study, in addition to the situation that the comparisons were performed with the alternative or estimated values.

Nevertheless, DSF using T1rLBD is expected to serve as an effective screening method to find chemicals potentially serving as taste substances for T1rs at the first stage of research, followed by further analyses to clarify their actual activities. Since the binding manner of taste substances at the orthosteric site in LBD is likely common to T1rs, the method may be useful for sweet or umami substance screening if recombinant protein preparation of human T1rLBD is achieved in future.

## Supporting information

**S1 Fig. Confirmation of the thermal stabilization of T1r2a/T1r3LBD in 20 mM HEPES-- NaOH, 300 mM NaCl, 2 mM CaCl$_2$, pH 7.5.** The average $\Delta T_m$ of T1r2a/T1r3LBD in the presence of 0.1, 1, and 10 mM of L-glutamine, L-alanine, and D-alanine are shown. Error bars, s.e. m. ($n = 4$). The $T_0$ in this condition was determined as 56.3 $\pm$ 0.06 ˚C ($n = 12$). Please also see Fig 3A.
(PDF)

**S2 Fig. Thermal melt curves of T1r2a/T1r3LBD (top) and their derivatives (bottom) in the presence of 0.1, 1, and 10 mM of amino acids measured by DSF.**
(PDF)

**S3 Fig. Correlation between the DSF results and the response assay results.** (A) The thermal stabilization of LBD in the presence of 10 mM of amino acid, shown in $\Delta T_m$, is plotted on the full-length receptor responses to the same concentration of amino acid, shown in $\Delta$RFU. (B) The affinities to the LBD estimated by the DSF ($pK_{d\text{-app}} = \log 1/K_{d\text{-app}}$) were plotted on the estimated amino acid potencies for the receptor activation ($pEC_{50\text{-est}} = \log 1/EC_{50\text{-est}}$, estimated from the responses at 10 mM concentration). (C) The affinities to the LBD estimated by the DSF ($pK_{d\text{-app}}$) were plotted on the estimated amino acid potencies for the receptor activation ($pEC_{50\text{-est}}$, estimated from the responses at 5 mM concentration).
(PDF)

**S1 Table. $\Delta T_m$ and derived $K_{d\text{-app}}$ values estimated from the DSF results of T1r2a/T13LBD at a single ligand concentration.**
(PDF)

**S2 Table. $\Delta$RFU and derived $EC_{50\text{-est}}$ values derived from the response assay of T1r2a/T13.**
(PDF)

**S3 Table. Affinities to mfT1r2a/T1r3LBD derived from the DSF results and physicochemical parameters for the $\alpha$-substituent group of amino acids.**
(PDF)

## Acknowledgments

We thank Dr. Harumi Fukada for her advice on data analysis and the manuscript.

## Author Contributions

**Conceptualization:** Takashi Yoshida, Norihisa Yasui, Atsuko Yamashita.

**Formal analysis:** Takashi Yoshida, Yuko Kusakabe, Chiaki Ito, Miki Akamatsu, Atsuko Yamashita.

**Funding acquisition:** Norihisa Yasui, Atsuko Yamashita.

**Investigation:** Takashi Yoshida, Norihisa Yasui, Yuko Kusakabe, Chiaki Ito.

**Writing – original draft:** Takashi Yoshida, Atsuko Yamashita.

**Writing – review & editing:** Norihisa Yasui, Yuko Kusakabe, Miki Akamatsu.

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
