## [Decision Letter · Decision Letter 0]

21 Aug 2019

PONE-D-19-16465

Differential scanning fluorimetric analysis of the amino-acid binding to taste receptor using a model receptor protein, the ligand-binding domain of fish T1r2a/T1r3

PLOS ONE

Dear Dr. Yamashita,

Thank you for submitting your manuscript to PLOS ONE. After careful consideration, we feel that it has merit but does not fully meet PLOS ONE’s publication criteria as it currently stands. Therefore, we invite you to submit a revised version of the manuscript that addresses the points raised during the review process.

We would appreciate receiving your revised manuscript by Oct 05 2019 11:59PM. To enhance the reproducibility of your results, we recommend that if applicable you deposit your laboratory protocols in protocols.io, where a protocol can be assigned its own identifier (DOI) such that it can be cited independently in the future. For instructions see: http://journals.plos.org/plosone/s/submission-guidelines#loc-laboratory-protocols

We look forward to receiving your revised manuscript.

Kind regards,

Piero Andrea Temussi

Academic Editor

PLOS ONE

Journal Requirements:

Reviewers' comments:

Reviewer's Responses to Questions

**Comments to the Author**

1. Is the manuscript technically sound, and do the data support the conclusions?

Reviewer #1: Yes

2. Has the statistical analysis been performed appropriately and rigorously? 

Reviewer #1: N/A

3. Have the authors made all data underlying the findings in their manuscript fully available?

Reviewer #1: Yes

4. Is the manuscript presented in an intelligible fashion and written in standard English?

Reviewer #1: Yes

5. Review Comments to the Author

Reviewer #1: The manuscript "Differential scanning fluorimetric analysis of the amino-acid binding to taste receptor using a model receptor protein, the ligand-binding domain of fish T1r2a/T1r3" presents interactions between various amino acids and taste receptor by differential scanning fluorimetry.

This work is another important step how amino acids interact with taste receptor. However, the lack of detailed conditions in DSF analysis, the reviewer concluded this manuscript is not recommended to publish in the present form. This manuscript would be strengthened by a more careful and detailed description of the methods as well as results as explained below.

As shown in the Figure 1A, two potential binding sites for amino acids seem to exist in the receptor, however, no detailed comments are presented from DSF results. The amino acids ligands truly bind two sites? The authors should be clarified the binding specificity of ligands as to each site including DSF results.

In Figure 1B (Gln), and Figure 2A (Ala, Glu, Arg), one minor peak, around 50�C is visible.

What is this minor peak? As the peak is more obvious as ligands concentration increases, some other effects such as a pH change might be involved. The authors should be clearly addressed these issues.

Several other points that should be addressed:

In DSF experiments, the authors use Tris buffer. Generally, this buffer is suspected to induce pH change when temperature increases. The reviewer thinks DSF results also might be influenced by pH change as Tm values also seem to be sensitive to type of amino acids. I could not judge from the present manuscript, though, the pH value with each ligand, especially at high concentration, should be indicated in the manuscript.

Figure 2B

Label typo for Glu

The reviewer also thinks this manuscript would become more attractive if possible negative controls were included (amino acids ligand which does not bind or weakly binds receptor since high concentration amino acids ligands seems to be involved in non-specific binding to receptor.

6. PLOS authors have the option to publish the peer review history of their article (what does this mean?). If published, this will include your full peer review and any attached files.

Reviewer #1: No

---

## [Author Response · Author response to Decision Letter 0]

2 Sep 2019

Responses to Reviewer

Reviewer #1:

The manuscript "Differential scanning fluorimetric analysis of the amino-acid binding to taste receptor using a model receptor protein, the ligand-binding domain of fish T1r2a/T1r3" presents interactions between various amino acids and taste receptor by differential scanning fluorimetry.

This work is another important step how amino acids interact with taste receptor. However, the lack of detailed conditions in DSF analysis, the reviewer concluded this manuscript is not recommended to publish in the present form. This manuscript would be strengthened by a more careful and detailed description of the methods as well as results as explained below.

Reply: We appreciate the reviewer’s valuable comments on our manuscript. We added descriptions in the sections pointed by the reviewer, as described below.

As shown in the Figure 1A, two potential binding sites for amino acids seem to exist in the receptor, however, no detailed comments are presented from DSF results. The amino acids ligands truly bind two sites? The authors should be clarified the binding specificity of ligands as to each site including DSF results.

Reply: We added the discussion about the presence of two potential binding sites in the protein and the DSF results in this study at l. 370-374, p. 17.

In Figure 1B (Gln), and Figure 2A (Ala, Glu, Arg), one minor peak, around 50�C is visible.

What is this minor peak? As the peak is more obvious as ligands concentration increases, some other effects such as a pH change might be involved. The authors should be clearly addressed these issues.

Reply: We have a section discussing about the biphasic melting transition of the protein, indicated by the presence of two peaks (two maxima) in their derivatives, at “Thermodynamic properties of T1r2a/T1r3LBD” in Discussion. In the last paragraph of this section, we discussed about the melting transition around 50 ˚C, corresponding to the peak around 50 ˚C in the derivatives pointed by the reviewer. To make this more explicit, we added the notions correlating the melting transition with the maxima in their derivatives at l. 195-199, p. 9-10, and l. 345-346 in p. 16.

In DSF experiments, the authors use Tris buffer. Generally, this buffer is suspected to induce pH change when temperature increases. The reviewer thinks DSF results also might be influenced by pH change as Tm values also seem to be sensitive to type of amino acids. I could not judge from the present manuscript, though, the pH value with each ligand, especially at high concentration, should be indicated in the manuscript.

Reply: We confirmed that the use of HEPES buffer, one of the buffers with more stable pH upon temperature increase, gave consistent Tm shifts with those observed in this study for several representative amino acid ligands. We added this information at l.116-118, p. 6 and S1 Fig in the revised manuscript. S1 Fig and S2 Fig in the original manuscript were renamed as S2 Fig and S3 Fig accordingly.

For the experiments in this study, we adjusted the pH of the amino acid solution to ~ 8.0 before mixing with the protein, thus the pH change by the addition of the amino acid was unlikely. We added the detailed description of the preparation of the assay solution at l. 106-109, p. 6.

Figure 2B

Label typo for Glu

Reply: We corrected the label.

The reviewer also thinks this manuscript would become more attractive if possible negative controls were included (amino acids ligand which does not bind or weakly binds receptor since high concentration amino acids ligands seems to be involved in non-specific binding to receptor.

Reply: We already have a negative control, D-alanine, exhibiting no substantial Tm shifts by addition of up to 10 mM, the highest amino-acid concentration tested in this study, as shown in Fig. 3A and described at l.257-261, p. 12.

---

## [Editor Report · Decision Letter 1]

18 Sep 2019

Differential scanning fluorimetric analysis of the amino-acid binding to taste receptor using a model receptor protein, the ligand-binding domain of fish T1r2a/T1r3

PONE-D-19-16465R1

Dear Dr. Yamashita,

We are pleased to inform you that your manuscript has been judged scientifically suitable for publication and will be formally accepted for publication once it complies with all outstanding technical requirements.

With kind regards,

Piero Andrea Temussi

Academic Editor

PLOS ONE
---

## [Editor Report · Acceptance letter]

23 Sep 2019

PONE-D-19-16465R1 

Differential scanning fluorimetric analysis of the amino-acid binding to taste receptor using a model receptor protein, the ligand-binding domain of fish T1r2a/T1r3 

Dear Dr. Yamashita:

I am pleased to inform you that your manuscript has been deemed suitable for publication in PLOS ONE. Congratulations! Your manuscript is now with our production department. 

With kind regards,

on behalf of

Dr. Piero Andrea Temussi 

Academic Editor

PLOS ONE